# How Much Complexity Is Required for Modelling Grassland Production at Regional Scales?

Iris Vogeler [1,2,*], Christof Kluß [1], Tammo Peters [1] and Friedhelm Taube [1,3]

1   Grass and Forage Science/Organic Agriculture, Christian Albrechts University, 24118 Kiel, Germany
2   Department of Agroecology, Aarhus University, 8830 Tjele, Denmark
3   Grass Based Dairy Systems, Animal Production Systems Group, Wageningen University,
    6708 WD Wageningen, The Netherlands
*   Correspondence: ivogeler@gfo.uni-kiel.de

**Abstract:** Studies evaluating the complexity of models, which are suitable to simulate grass growth at regional scales in intensive grassland production systems are scarce. Therefore, two different grass growth models (GrasProg1.0 and APSIM) with different complexity and input requirements were compared against long-term observations from variety trials with perennial ryegrass (*Lolium perenne*) in Germany and Denmark. The trial sites covered a large range of environmental conditions, with annual average temperatures ranging from 5.9 to 10.3 °C, and annual rainfall from 536 to 1154 mm. The sites also varied regarding soil type, which were for modelling categorised into three different groups according to their plant available water (PAW) content: light soils with a PAW of 60 mm, medium soils with a PAW of 80 mm, and heavy soils with a PAW of 100 mm. The objective was to investigate whether the simple model performed equally well with the given low number of inputs, namely climate and PAW group. Evaluation statistics showed that both models provided satisfactory results, with root mean square errors for individual cuts ranging from 0.59 to 1.28 t dry matter ha$^{-1}$. The model efficiency (Nash–Sutcliffe efficiency) for the separate cuts were also good for both models, with 81% of the sites having a positive Nash–Sutcliffe efficiency value with GrasProg1.0, and 72% with APSIM. These results reveal that without detailed site-specific descriptions, the less complex GrasProg1.0 model can be incorporated into a simple decision support tool for optimising grassland management in intensive livestock production systems.

**Keywords:** GrasProg1.0; APSIM; perennial ryegrass; North-West Europe

## 1. Introduction

Despite the importance of grasslands in sustaining ruminant livestock farming, information about grassland productivity and its response to changing climatic conditions, with increasing frequency and severity of extreme events, is scarce [1–3]. Simulation models constitute a key tool to understanding and predicting the effects of climate variations and management strategies on biophysical systems. Various models have been developed and used for predicting grass growth. Modelling approaches vary from simple empirical to complex mechanistic models, and operate on different hierarchical levels, from the individual plant [4], to plant communities based on plant functional types [5,6], and to the field [7–10] or even global scale [11,12].

Complex process-based models at the individual plant level include numerous plant-physiological functions, which are very parameter intensive and data demanding [6,13]. For modelling at higher hierarchical levels, simple physiological and morphological plant traits as well as statistical functions, which represent dynamic plant growth processes, have been integrated into mechanistic models [7,10,14]. Some of these simpler dynamic and mechanistic modelling approaches have also been integrated into decision support tools for practical grassland management [15]. The compromise between model complexity and

input data requirement has been addressed in several studies, and the selection of the model for a given application depends, among other factors, on the expected performance, data availability as well as the users' familiarity with the model [16,17].

In Europe, grasslands amount to about 34% of the agricultural area [18], with a similar share in Germany of 28% [19]. Grasslands provide a wide range of ecosystem services including carbon sequestration, water filtering, and the provision of habitats for wildlife [20–22]. Apart from delivering substantial ecosystem services, grasslands are a low-cost feed source for ruminants. Perennial ryegrass (*Lolium perenne*) is the most important forage grass in temperate climates due to its high dry matter (DM) productivity potential in combination with a high forage digestibility and nutritive value throughout the grazing season [23]. However, temperature-limited herbage growth in spring and autumn and moisture-limited growth in summer can result in feed deficits in intensively managed systems. Thus, future grass growth and thus feed supply is highly uncertain within and between seasons and locations. Extreme drought periods have been shown to prolong the start of growth after rewetting, which has been referred to as the legacy effect. This can particularly occur in shallow-rooted grasses such as perennial ryegrass, which further influences the annual yield variability [24].

Due to climatic conditions, grass growth rates are highly variable both in time (within and between seasons at one location), and in space (between locations). For example, average annual dry matter (DM) yields of perennial ryegrass in Germany show substantial inter-annual variations as well as large variations within the various states of Germany, with annual DM yields in the last decade ranging from <3 t ha$^{-1}$ to >9 t ha$^{-1}$ (Figure 1). This is due to differences in the soils (including availability of water and nutrients) as well as the high temporal variability in weather conditions. For example, the extended summer drought all over North-West Europe in 2018 is reflected in a substantial drop in DM yield, with reductions ranging from 60 to 93% compared with the average of the last 10 years. This high variability has direct impacts on the levels of forage produced on farms, and thus the feeding management.

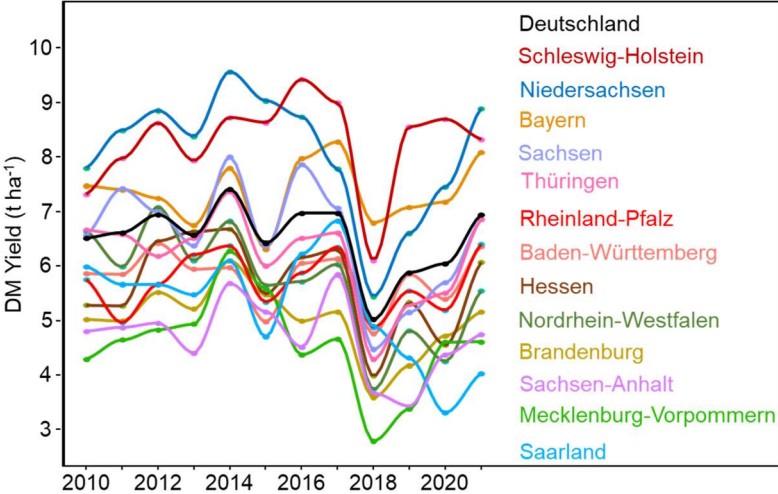

**Figure 1.** Annual dry matter (DM) yields for the different states of Germany obtained from the Statistisches Bundesamt (https://www.destatis.de/DE/Themen/Branchen-Unternehmen/Landw irtschaft-Forstwirtschaft-Fischerei/Publikationen/Bodennutzung/landwirtschaftliche-nutzflaeche -2030312217004.pdf; accessed 11 January 2021).

Many model comparison studies have been conducted for cropping systems, and the use of model outputs of model ensembles is gaining attention [25,26]. Only a few comparisons of grass growth models with different complexities (deterministic and empirical) have been conducted [27–29]. While these few studies suggest that empirical models are often comparable to more complex deterministic models, further evaluation

is required, especially with studies covering more diverse climatic conditions. Moreover, evaluating the model complexity (i.e., the number of variables) is of importance as the addition of new variables may increase the probability of additional errors and less accurate simulations [30,31].

In mechanistic and process-oriented modelling approaches such as the Agricultural Production Systems Simulator (APSIM; [32]), pools and fluxes of carbon, nitrogen, and water are represented, with sub-models for soil, water, carbon, and nitrogen, and the plant. Plant growth is calculated based on physiological processes at the plant scale (i.e., considering the leaf photosynthetic rate and carbon accumulation based on incoming radiation, carbon dioxide, temperature, water, nitrogen, plant respiration, fertility, and tissue turnover including senescence and detachment of dead material) [33]. In more simple semi-mechanistic models, detailed physiological processes are not considered in detail. In the GrasProg model, for example, the simulations are based on functional growth equations that consider more general physiological principles of perennial grasses. Here, the relative growth rate and development of the leaf area index as a function of time are integral parts of the model to account for the photosynthetic efficiency of the grassland canopy with subsequent considerations of the growth limiting factors for temperature, radiation, and precipitation [34].

The objective of the current study was to compare two models with different complexity, GrasProg1.0, an updated version of GrasProg [34] and APSIM, for predicting grass growth across Northern Europe using only basic soil information and typical fertilisation rates. The evaluation was based on long-term variety trials with perennial ryegrass from Germany and Denmark.

## 2. Materials and Methods

### 2.1. Trial Sites

For the comparison of the two models, the grass growth data from Germany and Denmark were used (Figure 2). In Germany, data were obtained from the states' variety testing trials (Landessortenversuche; sourced from http://www.landwirtschaftskammern.de/ accessed 12 April 2021), and in Denmark from the national trials recorded in the Nordic Field Trial System (NFTS; https://nfts.dlbr.dk/Forms/Forside.aspx; assessed 14 June 2021). These testing trials run over a period of three years, after which another set of new varieties is started. To represent the perennial character of permanent grassland and avoid the effects of poor grassland establishment in the first production year, data were limited to the second and third production/trial year. Additionally, only perennial ryegrass varieties from the medium maturity group (including reference varieties) were selected. These resulted in 28 sites for Germany and four for Denmark, spanning different soil types and a range of climatic conditions, with mean temperatures ranging from 5.9 to 10.3 °C and a mean annual rainfall from 536 to 1154 mm (Table 1).

According to the protocol in the trials from Germany, nitrogen (N) fertilisation ranged between 300 and 360 kg N ha$^{-1}$, of which 80–100 kg ha$^{-1}$ was applied in early spring for the first cut, which, according to the prescribed management protocols [37], should be carried out at BBCH51 (Biologische Bundesanstalt für Land- und Forstwirtschaft, Bundessortenamt und CHemische Industrie). For further details, see [38]. Depending on the site, cuts were taken in an area of 10 to 12 m$^2$, with four replications and cut to a height of 5–6 cm above ground. The DM content of the herbage was determined after oven drying the subsamples at 60 °C for 48 h. In Denmark, N fertilisation was applied according to the Danish Plant Directorates standards of 340 kg N ha$^{-1}$ yr$^{-1}$ for a pure grass, with 40% applied in early spring, 30% after the first harvest, 20% after the second harvest, and 10% after the third harvest. The plot size was 18 m$^2$, and the cutting height was 5 cm. Dry matter was determined by oven drying the subsamples at 60 °C for 40 h.

For the modelling, the soils were categorised into three different groups based on their plant available water content (PAW) in the rootzone, namely 'low' with 60 mm, 'medium' with 80 mm, and 'high' with 100 mm (with a rootzone depth for ryegrass set as 500 mm).

For Germany, the PAW for the soils was derived from the Ackerzahl, which is based on the German land appraisal system (Reichsbodenschätzung), which was initiated in 1932 to rank soils according to their potential productivity [35]. The Ackerzahl is scaled from 1 to 100 (for highest productivity), and takes the soil type, formation, topography, and climatic conditions into account. Soils with an Ackerzahl up to 25 were classified into the 'low' PAW soil group, those with an Ackerzahl 25 and 65 to the 'medium' soil group, and those with an Ackerzahl larger than 65 to the 'high' soil group. For the Danish sites, soils were grouped according to the Danish soil classification scheme [36], with JB1 (coarse sandy) and JB2 (fine sandy soil) in the 'low' soil group, and JB5 and JB6 (sandy clay) in the 'high' soil group (Table 1).

Daily weather data were gathered from meteorological sites close by the trial sites. For Germany, these were obtained from the Deutschen Wetterdienst (Germany's National Meteorological Service, DWD; https://www.dwd.de/; accessed 12 April 2021). For Denmark, they were obtained from the online database (http://agro-web01t.uni.au.dk/KlimaDB/; accessed 14 June 2021) managed by Aarhus University.

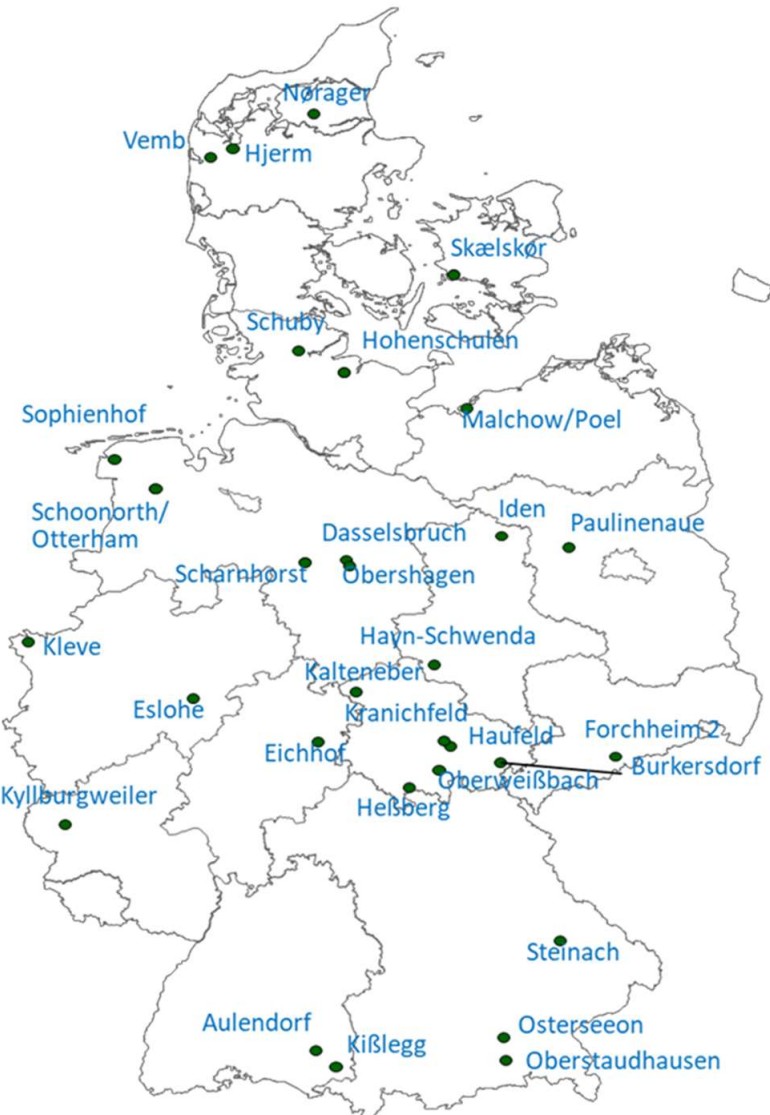

**Figure 2.** Locations of the national ryegrass variety trials in Germany and Denmark, used for the comparison of two models, GrasProg1.0 and APSIM, for the prediction of grass growth.

**Table 1.** Site descriptions and meteorological stations (Met Station) used for the modelling. DWD (=Germany's National Meteorological Service, Der Deutsche Wetterdienst) was used for the modelling. Lon = longitude, Lat = latitude in degrees, Alt = altitude in meters above sea level, PAW = plant available water (mm) in 500 mm depth, T = mean annual temperature (°C), mean annual RF = mean annual rainfall (mm), Lon Met = longitude in degrees of the meteorological station, Lat Met = latitude in degrees of the meteorological station, * proximate values. Soil classification (Soil) in Germany (DE) was based on the Ackerzahl [35] and on the Danish soil classification scheme [36] in Denmark (DK).

| Site | Lon | Lat | Alt | PAW | Soil | Met Station | Lon Met | Lat Met | T | RF |
|---|---|---|---|---|---|---|---|---|---|---|
| Aulendorf, DE | 9.66 | 47.94 | 570 | 80 | 56 | Weingarten | 9.62 | 47.81 | 9.3 | 926 |
| Burkersdorf, DE | 11.88 | 50.65 | 594 | 80 | 36 | Schleiz | 11.80 | 50.57 | 8.2 | 652 |
| Dasselsbruch, DE | 10.02 | 52.56 | 35 * | 60 | 20 | Celle | 10.03 | 52.60 | 10.0 | 679 |
| Eichhof, DE | 9.68 | 50.85 | 200 | 80 | 57 | Bad Hersfeld | 9.74 | 50.85 | 9.1 | 658 |
| Eslohe, DE | 8.17 | 51.25 | 370 * | 80 | 40 | Eslohe | 8.16 | 51.25 | 8.5 | 1086 |
| Forchheim 2, DE | 13.27 | 50.71 | 565 | 80 | 33 | Marienberg | 13.15 | 50.65 | 7.3 | 890 |
| Haufeld, DE | 11.28 | 50.80 | 430 | 80 | 56 | Jena | 11.58 | 50.93 | 10.3 | 594 |
| Hayn-Schwenda, DE | 11.08 | 51.57 | 441 | 80 | 40 | Harzgerode | 11.14 | 51.65 | 8.0 | 582 |
| Heßberg, DE | 10.78 | 50.42 | 380 | 80 | 45 | Lautertal | 10.97 | 50.31 | 9.1 | 739 |
| Hjerm, DK | 8.65 | 56.43 | 30 * | 100 | JB5/6 | Vemb | 8.22 | 56.71 | 8.6 | 796 |
| Hohenschulen, DE | 9.99 | 54.32 | 30 | 80 | 50 | Kiel-Holtenau | 10.14 | 54.38 | 9.4 | 759 |
| Iden, DE | 11.90 | 52.78 | 18 | 100 | 67 | Seehausen | 11.73 | 52.89 | 9.6 | 565 |
| Kalteneber, DE | 10.14 | 51.32 | 450 * | 80 | 45 | Leinefelde | 10.31 | 51.39 | 8.6 | 700 |
| Kißlegg, DE | 9.89 | 47.79 | 700 * | 80 | 58 | Weingarten | 9.62 | 47.81 | 9.3 | 926 |
| Kleve, DE | 6.17 | 51.79 | 15 | 80 | 56 | Kleve | 6.10 | 51.76 | 10.3 | 837 |
| Kranichfeld, DE | 11.20 | 50.86 | 330 * | 80 | 46 | Erfurt-Weimar | 10.96 | 50.98 | 9.0 | 536 |
| Kyllburgweiler, DE | 6.62 | 50.07 | 529 | 80 | 34 | Manderscheid | 6.80 | 50.10 | 8.6 | 887 |
| Malchow/Poel, DE | 11.47 | 53.99 | 10 * | 80 | 34 | Boltenhagen | 11.19 | 54.00 | 9.3 | 597 |
| Nørager, DK | 9.63 | 56.75 | 40 * | 60 | JB2 | Aars | 9.51 | 56.76 | 8.3 | 706 |
| Obershagen, DE | 10.06 | 52.50 | 40 * | 80 | 45 | Celle | 10.03 | 52.60 | 10.0 | 679 |
| Oberstaudhausen, DE | 11.95 | 47.86 | 500 * | 80 | | Rosenheim | 12.13 | 47.88 | 9.2 | 1068 |
| Oberweißbach, DE | 11.14 | 50.58 | 660 | 60 | 23 | Neuhaus | 11.13 | 50.50 | 5.9 | 1154 |
| Osterseeon, DE | 11.93 | 48.07 | 560 | 80 | 45 | Ebersberg | 11.99 | 48.10 | 8.7 | 1036 |
| Ovelgönne, DE | 8.42 | 53.34 | 0 * | 100 | 88 | Bremerhaven | 8.58 | 53.53 | 10.1 | 753 |
| Paulinenaue, DE | 12.71 | 52.67 | 30 * | 60 | 30 | Neuruppin | 12.85 | 52.94 | 9.6 | 620 |
| Scharnhorst, DE | 9.52 | 52.53 | 38 | 80 | 50 | Wunstorf | 9.43 | 52.46 | 10.3 | 650 |
| Schoonorth-Otterham, DE | 7.22 | 53.50 | −0.3 | 100 | 85 | Emden | 7.23 | 53.39 | 9.4 | 823 |
| Schuby, DE | 9.45 | 54.52 | 42.7 | 60 | 22 | Schleswig | 9.55 | 54.53 | 8.6 | 885 |
| Skælskør, DK | 11.31 | 55.24 | 6 * | 100 | JB6 | Flakkebjerg | 11.39 | 55.31 | 8.9 | 581 |
| Sophienhof, DE | 9.06 | 49.81 | 453 | 100 | 72 | Michelstadt-Vielbrunn | 9.10 | 49.72 | 8.5 | 1031 |
| Steinach, DE | 12.61 | 48.98 | 508 * | 80 | 56 | Straubing | 12.56 | 48.83 | 9.2 | 691 |
| Vemb, DK | 8.38 | 56.35 | 6 * | 60 | JB1/2 | Vemb | 8.22 | 56.71 | 8.6 | 796 |

## 2.2. Model Descriptions

### 2.2.1. GrasProg1.0

The GrasProg model is a semi-mechanistic model for simulating grass growth for intensively managed ryegrass (*Lolium perenne*) dominated swards with typical non limiting N fertilisation rates. The model only requires a few input parameters, and aside from proxies for the number of generative tillers and the tiller density, only the soil's plant

available water (PAW) and meteorological factors (global radiation, mean daily temperature, precipitation, and evaporation) are necessary. The model has previously been calibrated for intensively managed ryegrass dominated grass swards with typical non limiting N fertilisation rates in North-West Germany (GrasProg; [34]). Now included in the updated version, GrasProg1.0 is a drought legacy factor that accounts for a period of unusually dry weather. Such extreme and long drought events can, aside from an immediate reduction in canopy photosynthesis, have longer-lasting legacy effects on vegetation growth [39,40]. The drought legacy factor is assumed to start after a drought period of seven days, after which the start of the grass growth is delayed by 7 days, where a drought is defined as the soil having a water content ≤30% PAW.

The model was set up for the trial sites described above using the meteorological data from the climate stations nearby (Table 1) and the site relevant soil PAW, either low (PAW = 60 mm), medium (PAW = 80 mm), or high (PAW = 100 mm).

### 2.2.2. APSIM

APSIM is a modular process-oriented simulation framework maintained by the APSIM Initiative (www.apsim.info; accessed 14 June 2021). APSIM is climate-driven and comprises a range of submodels including SoilWat for simulating water movement, SoilNitrogen for simulating N cycling, AgPasture for pasture growth and N uptake, and the Micromet module [41] for computing evapotranspiration using the Penman–Monteith equation. AgPasture is based on the physiological model of Thornley and Johnson [42], which has been shown to simulate growth patterns and seasonal yields well [43,44]. In brief, grass growth is modelled with a daily time-step calculation based on intercepted global solar radiation, radiation use efficiency, and growth modifiers for temperature, soil water, and N supply. APSIM with the AgPasture model has been used successfully for simulating grass growth under a range of climatic conditions in New Zealand, mainly binary mixtures of ryegrass/white clover [45], but also for diverse pastures [46,47] and for annual and perennial ryegrass in Australia [48]. The model has also been tested for predicting seasonal grass growth rates under different climatic conditions for New Zealand and using generic soils with PAWs estimated from the land use capability classes [49].

The model was set up with a pure perennial ryegrass (*Lolium perenne* L.), a rooting depth of 500 mm, and three different soil profiles: light (PAW = 60 mm), medium (PAW = 80 mm), and heavy (PAW = 100 mm). The soil organic carbon in the top 100 mm was set according to averages for grassland and different soil types across Germany [50], with 3.8% for sandy soils (used for the light soils), 3.9 for loamy soils (used for the medium soils), and 2.9% for clay soils (used for the heavy soils). The grass was cut according to the trial management, and fertiliser was applied via a manager script, with the amounts and timings as described above. Meteorological data required by APSIM are daily values of rainfall, minimum and maximum daily temperature, and radiation.

### 2.2.3. Data Analysis and Statistical Analysis

Grass growth data were screened for outliers using the linear regression of pasture production of the first cut vs. global radiation sum and temperature sum from the beginning of the growing season (taken after a temperature sum of 250 °C with a base temperature of 3 °C) to the date of the first cut (Figure 3). Cook's distance, which measures the change in fitted response for all observations with and without the presence of observation i, was then used to identify outliers. Observations that have a Cook's distance >4 times the mean were classified as outliers.

The performance of GrasProg1.0 and APSIM were evaluated based on common measures including the coefficient of determination ($R^2$), Nash–Sutcliffe efficiency (NSE), root mean square error (RMSE), and percent bias. For these, the R package hydroGOF [51] was used. Additionally, a paired *t*-test was conducted using the R function: *t*-test (x, y, paired = TRUE, alternative = "two.sided").

These statistics were calculated for both the entire dataset and for the individual sites using data from each individual cut. For the evaluation of the two models, the biomass of individual cuts as well as the annual amounts were used.

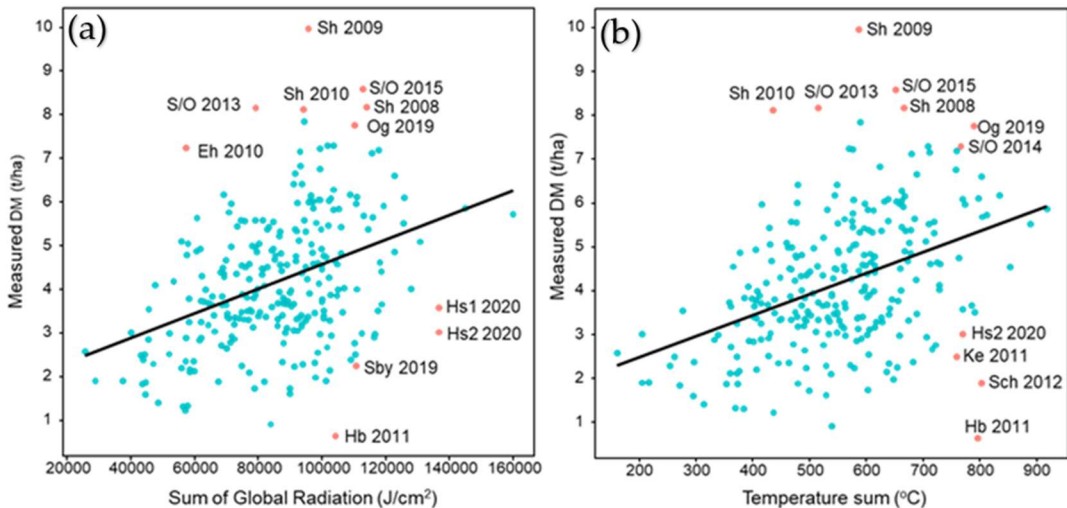

**Figure 3.** Measured grass dry matter (DM) yield of the first cut vs. temperature sum (**a**) and global radiation sum (**b**) after a temperature sum of 250 °C with a base temperature of 3 °C. Blue symbols indicated data that were included in the analysis, and red symbols those that were excluded for the years provided. Eh = Eichhof. Hb = Heßberg; Hs = Hohenschulen; Ke = Kalteneber; Og = Ovelgönne; Sby = Schuby, Sch = Scharnhorst; Sh = Sophienhof; S/O = Schoonorth/Otterham.

## 3. Results

### 3.1. Inclusion of a Legacy Effect

The improvement in GrasProg1.0 with the legacy effect can be seen in some of the data collected in 2018, which had a prolonged summer drought (Figure 4). While both versions of the model predicted the first cut in Kyllburgweiler well, the second and third were overestimated without the legacy effect. For Osterseeon, including the legacy effect reduced the grass growth during June too much, but in August, the simulations were much closer to the measurements. This shows that the inclusion of a legacy effect improved the model, but better parametrisation and/or its description in the model is required.

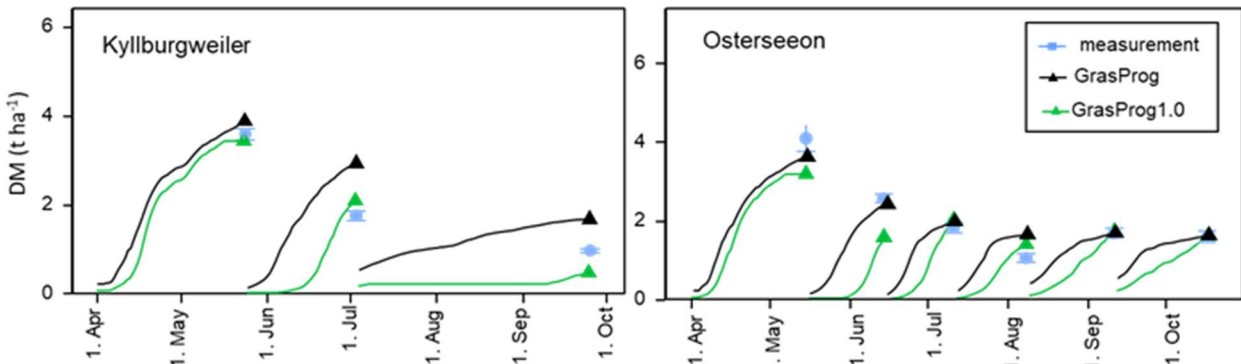

**Figure 4.** GrasProg1.0 predictions of the grass growth (dry matter; DM) in 2018 for Kyllburgweiler and Osterseeon, either without (black lines) or with the legacy (green lines) effect.

### 3.2. Measured and Predicted Dry Matter Production—Individual Cuts

Predictions of seasonal DM production (individual cuts) by GrasProg1.0 for four selected sites from varying geographical areas, altitudes, and with different meteorolog-

ical conditions (Aulendorf, Kleve, Osterseeon and Schuby with measurements over 11 to 12 years) showed generally good agreement with measurements (Figures 5–9). For GrasProg1.0, the RMSE ranged from 0.59 to 0.77 t ha$^{-1}$ and NSE from 0.55 to 0.71 for these four sites (Table 2; Figure 9). APSIM showed a slightly less good prediction for these four sites, with RMSE ranging from 0.59 to 0.91 t DM ha$^{-1}$ and NSE from 0.22 to 0.54. In some instances, GrasProg1.0 slightly underpredicted the first cuts while APSIM at times overpredicted these. The underestimation may be because GrasProg1.0 was calibrated on a dataset, which was more intensively defoliated (8 cuts yr$^{-1}$) compared with the data used for evaluation in the present study (4–5 cuts yr$^{-1}$). The defoliation frequency influences various plant traits such as tiller density, which greatly influence grass growth [52], which might explain the disparities between the measurements and simulations.

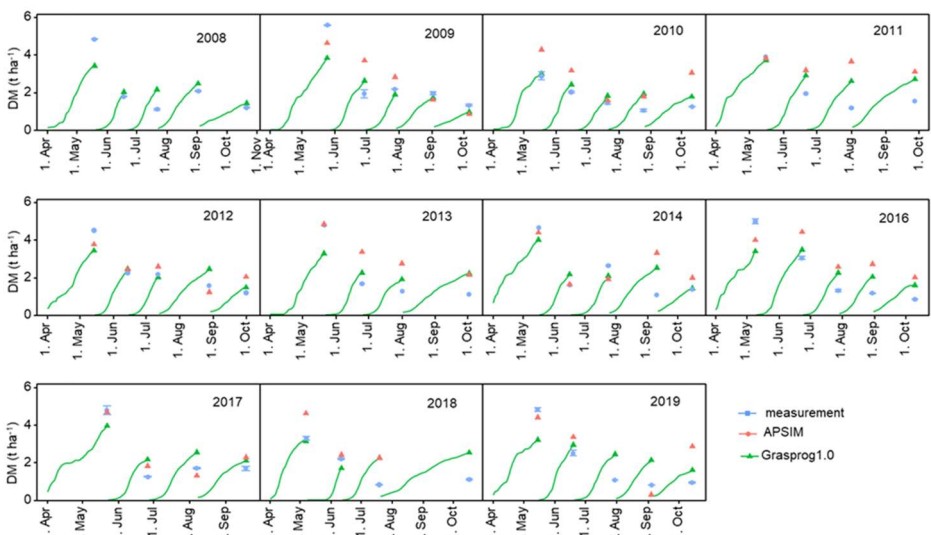

**Figure 5.** Measured grass dry matter (DM) for different cuts and years for Aulendorf, Germany. The predictions by GrasProg1.0 and APSIM are also shown.

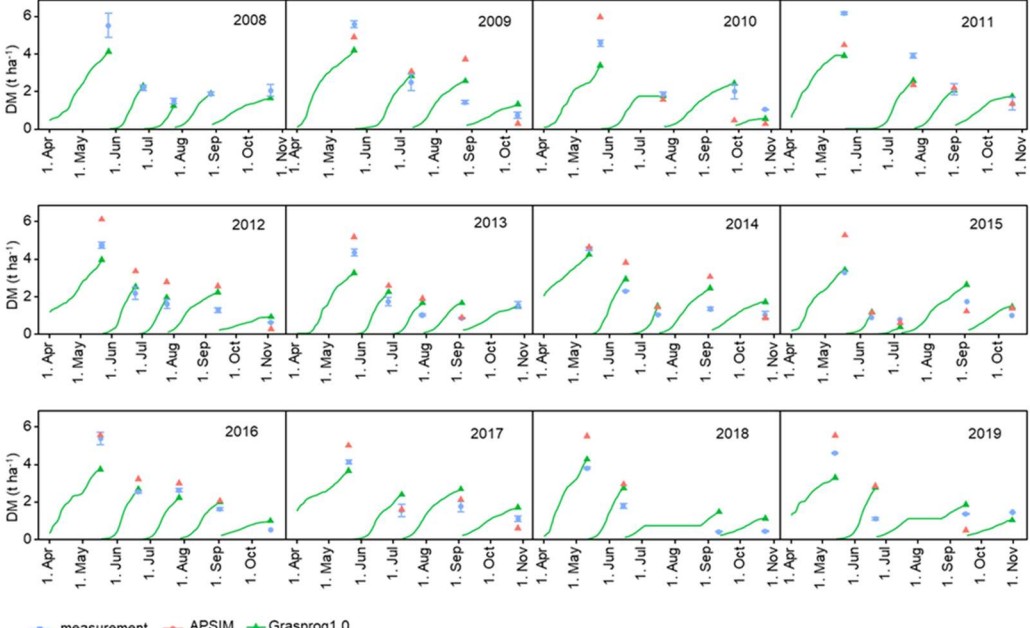

**Figure 6.** Measured grass dry matter (DM) for different cuts and years for Kleve, Germany. The predictions by GrasProg1.0 and APSIM are shown.

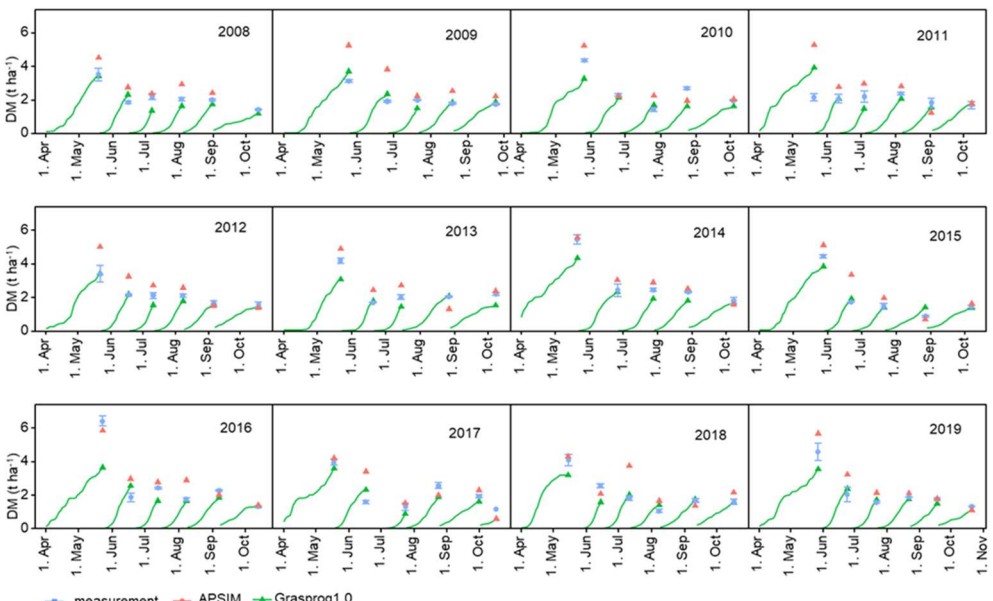

**Figure 7.** Measured grass dry matter (DM) for different cuts and years for Osterseeon, Germany. The predictions by GrasProg1.0 and APSIM are also shown.

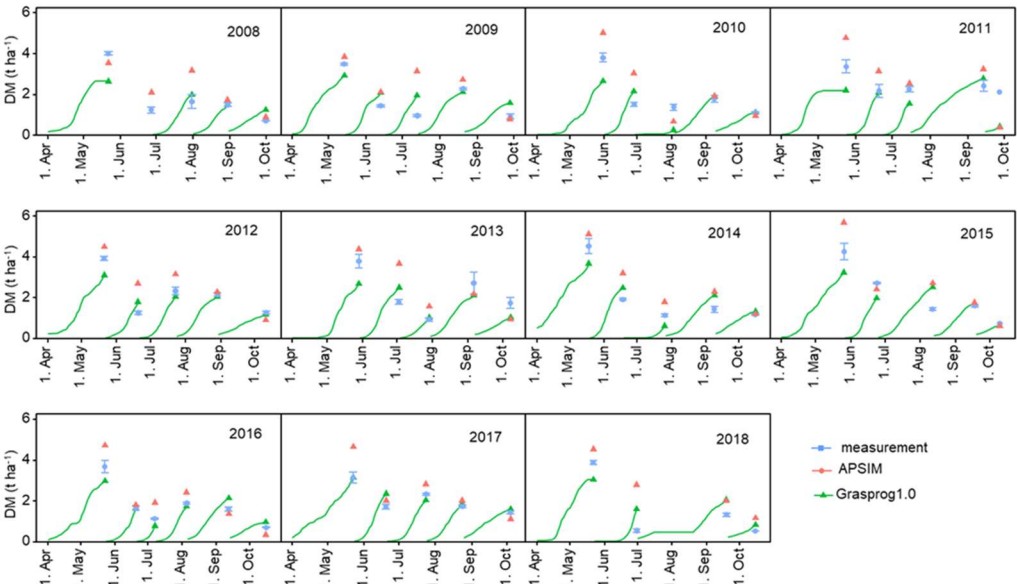

**Figure 8.** Measured grass dry matter (DM) for different cuts and years for Schuby, Germany. The predictions by GrasProg1.0 and APSIM are also shown.

For the entire datasets with a total of 32 different sites and measurement periods ranging 1 to 13 years, the RMSE values were acceptable, ranging from 0.59 to 1.28 t DM ha$^{-1}$ for GrasProg1.0 and 0.56 to 1.26 t ha$^{-1}$ for APSIM (Table 2). Accurately predicting grass growth with its high seasonal and interannual variation is not an easy task [28,53]. These RMSE values are lower to those reported with ranges from 0.7 to 2.1 t DM ha$^{-1}$, using three different models for predicting the first two cuts of timothy grass under northern European conditions [29]. They concluded that there is a need for a better understanding of the processes involved and how they are described in models. However, for some sites, the individual cuts were vastly over- or underpredicted, and the NSE values are close to zero or negative.

While there are no explicit standards for evaluating model performance, we used the suggested thresholds for monthly values to judge if the model results were satisfactory, namely NSE > 0.3, $R^2$ and *p*-value > 0.025 [54]. Out of the 32 datasets, GrasProg1.0 predicted grass growth for individual cuts satisfactorily for 11 datasets according to these criteria, and APSIM only slightly more with 13 datasets. Closer inspections also showed that GrasProg1.0 did not predict grass growth for the sites in Denmark satisfactorily, with high underestimation (high Pbias). This is not astonishing, as the model has not been calibrated for high latitude sites with long day-lengths. The APSIM model seemed to capture this slightly better, with two of the sites being satisfactorily simulated. Many of the datasets that were not satisfactorily predicted also had very short observation periods of ≥5 years.

Looking at the NSE values, GrasProg1.0 predicted the pattern of grass growth (individual cuts) on 81% of the sites better than just using the average values. When considering only observations ≥5 years, 85% of the sites were better predicted than using the averages.

Over the entire datasets, the individual cuts were reasonably predicted with GrasProg1.0, with a RMSE of 0.94 t DM ha$^{-1}$ and NSE of 0.43, and an overprediction of 12.9% (Figure 10). The performance of APSIM was slightly worse, with a RMSE of 0.99 t DM ha$^{-1}$, a NSE of 0.29, and an underprediction of 12.8%.

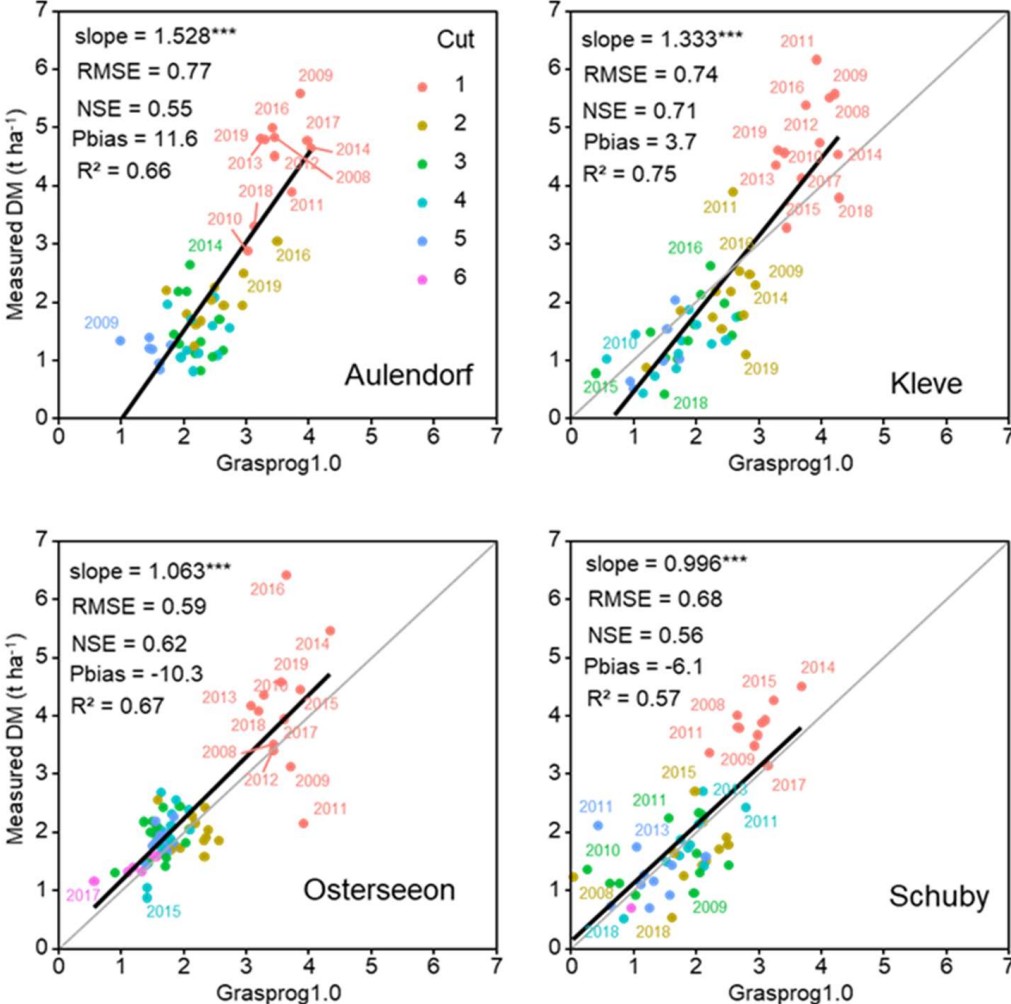

**Figure 9.** Measured grass dry matter (DM) vs. predictions by GrasProg1.0 for different cuts and years for Aulendorf, Kleve, Osterseeon, and Schuby, Germany. *** indicates significance *p* < 0.001.

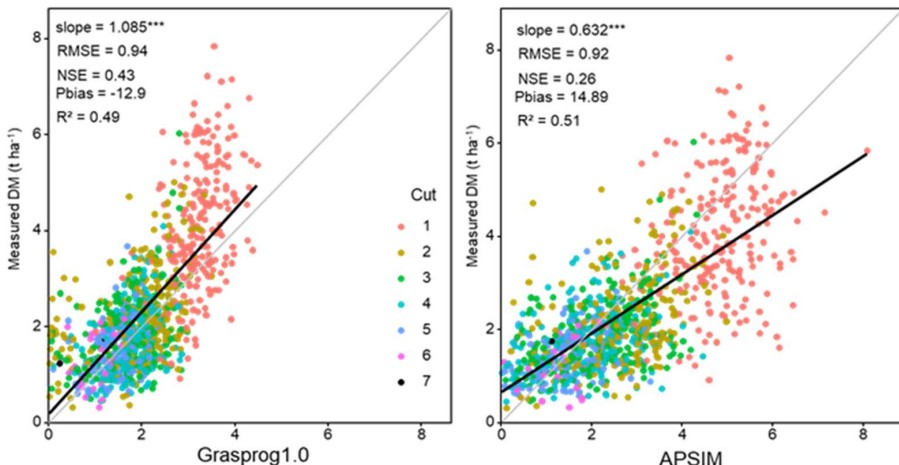

**Figure 10.** Measured grass dry matter (DM) vs. the predictions by GrasProg1.0 and APSIM for different cuts and years for the entire datasets from Germany and Denmark, with a total of 33 locations. *** indicates significance *p* < 0.001.

### 3.3. Measured and Predicted Dry Matter Production—Annual

Considering that the models were not tuned to any of the data at each site and the weather data were not obtained directly on the sites, both models performed well with a RMSE ranging from 0.12 to 2.85 t DM ha$^{-1}$ for GrasProg1.0 and from 0.06 to 2.55 t DM ha$^{-1}$ for APSIM. The inter-annual variability in annual yield was well-reflected (Figure 11). Furthermore, GrasProg was calibrated with a dataset from permanent grasslands, in which perennial ryegrass was the dominant species, but in which other grasses were also present. For accurate predictions, it has been suggested that the genetic variability between cultivars should be accounted for [29]. In our study, data from the national trials were restricted to the medium maturity group but comprised different cultivars. Furthermore, to increase the accuracy of the model for simulating growth in spring, the soil temperature should be considered, rather than the air temperature [34].

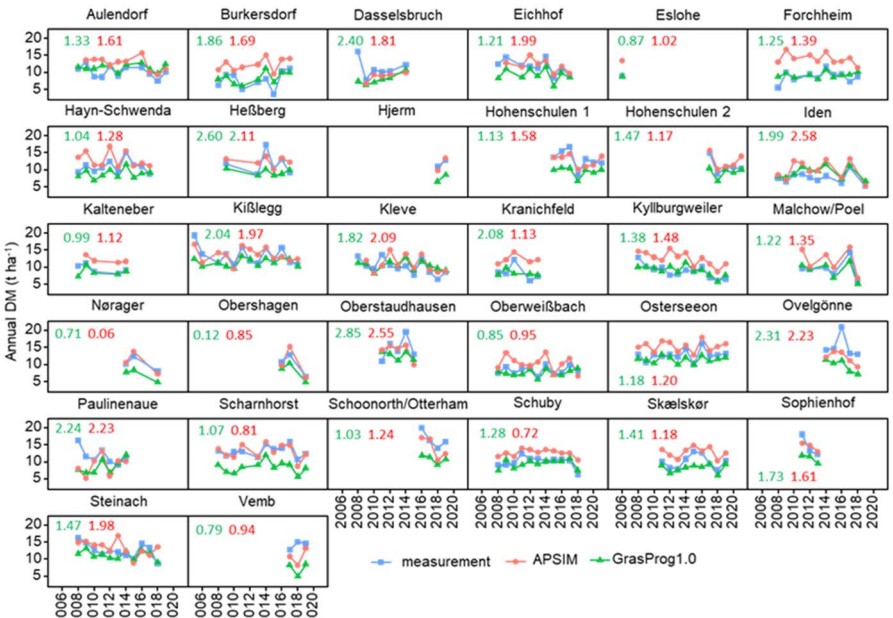

**Figure 11.** Annual measured grass dry matter (DM) and predictions by GrasProg1.0 and APSIM for different years for different locations in Germany and Denmark. The numbers show the RMSE for the two models (GrasProg1.0 green, APSIM red).

**Table 2.** Model performance statistics GrasProg1.0 and APSIM for individual cuts with data from variety trials from national trials in Germany and Denmark, spanning different numbers of years. Bold numbers indicate good model performance (NSE > 0.3, R² > 0.5, and *p* > 0.025), and x indicates if the model satisfies all three criteria.

| Site | RMSE | | R² | | NSE | | P (Paired *t*-Test) | | Pbias | | Slope | | Years | Evaluation | |
|---|---|---|---|---|---|---|---|---|---|---|---|---|---|---|---|
| | GrasProg | APSIM | GrasProg | APSIM | GrasProg | APSIM | GrasProg | APSIM | GrasProg | APSIM | GrasProg | APSIM | | GrasProg | APSIM |
| Iden | 0.72 | 0.49 | **0.68** | **0.76** | **0.64** | 0.10 | **0.094** | **0.555** | 12.40 | 30.60 | 1.10 | 0.57 | 14 | x | |
| Kißlegg | 0.69 | 0.70 | **0.55** | **0.55** | **0.43** | **0.40** | 0.009 | **0.649** | −14.00 | 2.00 | 0.98 | 0.65 | 13 | | x |
| Hayn-Schwenda | 0.92 | 0.76 | 0.42 | **0.60** | **0.32** | −0.24 | 0.000 | 0.001 | −18.50 | 19.70 | 0.99 | 0.48 | 12 | | |
| Kleve | 0.74 | 0.79 | **0.75** | **0.72** | **0.71** | **0.54** | **0.509** | **0.984** | 3.70 | 11.90 | 1.33 | 0.69 | 12 | x | x |
| Kyllburgweiler | 0.68 | 0.73 | **0.56** | 0.49 | **0.56** | −0.45 | **0.600** | 0.000 | 2.60 | 43.10 | 1.01 | 0.60 | 12 | x | |
| Osterseeon | 0.59 | 0.59 | **0.67** | **0.67** | **0.62** | 0.29 | 0.004 | 0.000 | −10.30 | 21.00 | 1.06 | 0.67 | 12 | | |
| Aulendorf | 0.77 | 0.91 | **0.66** | **0.52** | **0.55** | **0.31** | 0.019 | **0.374** | 11.60 | 24.80 | 1.53 | 0.77 | 11 | | x |
| Oberweißbach | 0.80 | 0.70 | 0.06 | 0.28 | −0.10 | −1.86 | 0.015 | 0.006 | −10.20 | 23.50 | 0.46 | 0.29 | 11 | | |
| Scharnhorst | 1.01 | 0.85 | **0.67** | **0.77** | **0.36** | **0.77** | 0.000 | **0.779** | −35.50 | −0.80 | 1.35 | 1.06 | 11 | | x |
| Schuby | 0.68 | 0.60 | **0.57** | **0.66** | **0.56** | 0.22 | **0.187** | 0.000 | −6.10 | 25.50 | 1.00 | 0.64 | 11 | x | |
| Steinach | 0.80 | 0.95 | 0.38 | 0.13 | 0.29 | −0.57 | 0.021 | **0.300** | −11.80 | 6.70 | 0.77 | 0.31 | 11 | | |
| Forchheim | 0.60 | 0.56 | 0.37 | 0.44 | **0.36** | −2.65 | **0.349** | 0.000 | 4.50 | 59.50 | 0.86 | 0.43 | 10 | | |
| Burkersdorf | 1.01 | 0.75 | **0.53** | **0.74** | **0.49** | 0.01 | **0.382** | 0.000 | 7.90 | 58.10 | 1.38 | 0.89 | 9 | x | |
| Eichhof | 0.98 | 0.73 | **0.59** | **0.73** | **0.48** | **0.63** | 0.000 | **0.538** | −20.70 | 3.90 | 1.22 | 0.73 | 9 | | x |
| Skælskør | 0.96 | 0.82 | 0.39 | **0.56** | 0.29 | 0.11 | 0.007 | 0.000 | −18.10 | 27.80 | 0.97 | 0.64 | 9 | | |
| Hohenschulen 1 | 0.85 | 0.81 | **0.66** | **0.69** | 0.25 | **0.69** | 0.001 | **0.513** | −27.40 | −3.50 | 1.55 | 0.95 | 7 | | x |
| Paulinenaue | 1.30 | 1.26 | 0.10 | 0.15 | −0.15 | −0.30 | **0.042** | **0.088** | −26.50 | −23.40 | 0.58 | 0.43 | 7 | | |
| Dasselsbruch | 1.21 | 0.93 | 0.16 | 0.21 | −0.25 | −0.19 | **0.032** | **0.173** | −29.40 | −14.10 | 0.63 | 0.47 | 6 | | |
| Eslohe | 0.68 | 0.57 | **0.53** | **0.66** | **0.54** | 0.15 | **0.378** | 0.002 | −4.20 | 28.00 | 1.08 | 0.77 | 6 | x | |
| Heßberg | 0.85 | 0.82 | **0.53** | **0.57** | **0.38** | −0.10 | **0.100** | **0.459** | −20.40 | 6.90 | 0.97 | 0.48 | 6 | x | |
| Malchow/Poel | 0.92 | 0.69 | **0.51** | **0.73** | **0.46** | **0.49** | **0.479** | 0.023 | −9.50 | 26.80 | 1.41 | 0.78 | 6 | x | |
| Hohenschulen 2 | 0.95 | 0.74 | **0.60** | **0.76** | **0.48** | **0.73** | **0.064** | **0.087** | −16.80 | 12.50 | 1.42 | 1.02 | 5 | x | x |
| Kalteneber | 0.90 | 0.81 | **0.56** | **0.58** | **0.55** | 0.09 | **0.196** | **0.912** | −9.40 | 32.30 | 1.16 | 0.68 | 5 | x | |
| Kranichfeld | 0.81 | 0.80 | **0.52** | **0.52** | **0.49** | −0.38 | **0.879** | 0.004 | −9.60 | 44.70 | 0.81 | 0.55 | 5 | x | |

**Table 2.** *Cont.*

| Site | RMSE | | R² | | NSE | | P (Paired *t*-Test) | | Pbias | | Slope | | Years | Evaluation | |
|---|---|---|---|---|---|---|---|---|---|---|---|---|---|---|---|
| | GrasProg | APSIM | GrasProg | APSIM | GrasProg | APSIM | GrasProg | APSIM | GrasProg | APSIM | GrasProg | APSIM | | GrasProg | APSIM |
| Oberstaudhausen | 0.70 | 0.75 | **0.43** | 0.36 | 0.24 | 0.10 | **0.216** | **0.638** | −17.40 | −3.30 | 0.91 | 0.54 | 5 | | |
| Ovelgönne | 0.95 | 0.94 | **0.66** | **0.67** | −0.08 | **0.46** | 0.008 | **0.047** | −37.30 | −21.00 | 1.38 | 0.96 | 5 | | x |
| Schoonorth | 0.74 | 0.72 | **0.75** | **0.76** | 0.01 | **0.66** | 0.005 | **0.087** | −34.90 | −14.40 | 1.63 | 1.01 | 4 | | x |
| Nørager | 0.73 | 0.45 | **0.56** | **0.83** | 0.05 | **0.53** | 0.024 | **0.593** | −31.70 | 4.00 | 0.98 | 0.62 | 3 | | x |
| Obershagen | 0.39 | 0.52 | **0.83** | 0.71 | **0.60** | 0.33 | 0.012 | **0.701** | −20.20 | 4.60 | 1.00 | 0.58 | 3 | | x |
| Sophienhof | 0.97 | 0.82 | **0.71** | **0.79** | 0.40 | **0.78** | **0.119** | **0.964** | −24.30 | −0.50 | 2.03 | 1.23 | 3 | | x |
| Vemb | 1.01 | 1.02 | 0.00 | 0.00 | −3.34 | −1.89 | **0.050** | **0.176** | −48.70 | -23.90 | −0.33 | 0.15 | 3 | | |
| Hjerm | 1.28 | 1.10 | 0.33 | **0.51** | −0.04 | **0.56** | 0.000 | **0.821** | −36.70 | -2.10 | 1.60 | 0.84 | 2 | | x |

## 4. Discussion

In practical grassland farming, information about yield often remains unknown or is only roughly estimated because management in recent decades has been animal-focused, rather than on grassland management [55]. The good prediction of grass growth by GrasProg1.0 means that the model can be used to aid farm management, especially by providing information on the best cutting or grazing times dependent on the climatic conditions. The model can also be used to evaluate likely changes due to climate change such as how increasing temperatures and temperature sums affect the cutting regimes and grassland productivity. The inclusion of the legacy factor in Grasprog1.0 means that the model can be employed to aid in the development of farm-level adaptations according to changes in the productivity and seasonality of grasslands resulting from the expected increases in drought and heat extremes [56]. Furthermore, model-based knowledge of annual yields can help to assist in optimising fertilisation strategies and avoid the risks of negative environmental effects due to N losses [57]. Knowledge about annual yields is also crucial, because according to the new German fertiliser ordinance [58], the permitted amount of N fertilisation needs to account for the yield of the grassland in previous years.

Trade-offs between model complexity and validation have been discussed [59] and include a more complete entity representation by complex models at the cost for the need of a greater requirement for validation and meta data. In contrast, simple models require less validation data, and model parameters are more generalised, with a greater probability of a large difference between the observed and estimated values. The APSIM model was not tuned to any of the data at each site, and general data were used rather than site-specific values such as soil hydraulic properties and organic carbon. The importance of accurate soil parameterisation when using a complex model such as APSIM for simulating soil water and nitrogen dynamics, and pasture production has also been emphasised by Craig et al. [60].

The similar fit between measurements and predictions by the two models means that GrasProg1.0 is a suitable grass growth model for North-West European conditions, especially where site-specific parameters are not available. This finding is in line with other studies where Hurtado-Uria et al. [28] found that an empirical model performed equally well as a complex model for predicting grass growth across Ireland, and Skinner et al. [27] also found no difference in the ability to simulate the grass forage yield in Pennsylvania, USA. The authors highlighted the need for better validation datasets for a robust comparison and parameterisation of the models. In contrast to other grass growth models such as GrazeGro [15], STICS [9], LINGRA [61], and even the simpler MoSt GG model [8], GrasProg only requires a few input parameters (namely temperature, radiation, rainfall, evaporation, and PAW soil group), and thus can be incorporated into a simple decision support tool for use by farmers and advisers.

When evaluating models, uncertainties in the observed data and the exact management of the sites should also be considered. The high spatial heterogeneity of botanical composition, nutrient availability, and defoliation strategy influence the forage biomass and quality, especially in permanent grasslands [62,63]. Additionally, due to the extreme variability of individual plants even at a small scale, the determination of grass biomass is very difficult [64], and the methodology of measurements influences the data of biomass and quality [65,66].

Furthermore, although the data used for model evaluation in this study were from national trials with prescribed management protocols, the specific management varied slightly between sites and years including differences in the amount of N applied and cutting regimes. However, the defoliation frequency can influence various plant traits such as the tiller density, which greatly influence grass growth [52] and may lead to additional disparities between the measurements and simulations. One limitation of GrasProg1.0 is that it does not account for N fertilisation management and currently does not include pasture quality indicators. This is mainly due to a lack of sufficient forage quality data across regions with different climatic conditions, covering seasonal pasture growth data under a range of fertiliser treatments. However, due to its generic structure, GrasProg1.0

has been shown to be suitable to deliver the information of biomass yields for intensively managed and perennial ryegrass dominated grasslands in Germany and northern Europe.

## 5. Conclusions

The hierarchical (plant, leaf, molecular) and spatial level (field, farm, landscape) at which grass growth is simulated is strongly dependent on the subsequent practical implementation of a model. For implementation as a decision support tool for grassland management, the simple semi-mechanistic model GrasProg1.0 is highly suitable and showed similar results to the more complex and process-oriented model APSIM. Such complex models are very data rich and require site-specific input parameters, which are often not known. In contrast, GrasProg1.0 only requires a few input parameters including meteorological data and the classification of the soil into a PAW class.

**Author Contributions:** Conceptualisation, F.T. and T.P.; Methodology, F.T. and T.P.; Software, C.K.; Validation, C.K., T.P. and I.V.; Formal analysis, C.K., T.P. and I.V.; Writing—original draft preparation, T.P. and I.V.; Writing—review and editing, I.V., T.P. and F.T.; Visualisation, C.K.; Supervision, F.T.; Funding acquisition, F.T. All authors have read and agreed to the published version of the manuscript.

**Funding:** This research was funded by the arismo GmbH within the project "GrasProg (Ertragsmodell Grünlandversicherung)".

**Institutional Review Board Statement:** Not applicable.

**Informed Consent Statement:** Not applicable.

**Data Availability Statement:** Data will be made available upon request.

**Acknowledgments:** We would like to thank Ingrid Nöhles and Regina Rehermann (Vereinigte Hagelversicherung, VVaG) and the arismo GmbH within the project "GrasProg (Ertragsmodell Grünlandversicherung)". We further would like to thank Mathias Herbst and Cathleen Frühauf (Deutscher Wetterdienst; DWD—Zentrum für Agrarmeteorologische Forschung Braunschweig; ZAMF) for weather data provision, and the institute managers of the "Landesdienststellen für Landwirtschaft" (state offices of agriculture) for the access to the national variety trials data, and special thanks to Christine Kalzendorf and Hubert Kivelitz for valuable discussion about the experimental setup and the analysis of modelling results.

**Conflicts of Interest:** The authors declare no conflict of interest.

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
