# Peer review of "How Much Complexity Is Required for Modelling Grassland Production at Regional Scales?"

_land, doi:10.3390/land12020327_

Round 1

Reviewer 1 Report

the article can be accepted in present form 

Author Response

Thank you for the positive evaulation

Reviewer 2 Report

The models in agriculture are useful tools in order to manage grassland ecosystems effectively. They are also important for predictions about the development of grass systems in time and space. This article discusses the ability of two modelling software to do so. Both are applied in the practice, but it seems that the authors need to prove that the one is better for certain applications. The study is of interest but by my opinion it needs some clarifications and improvements. Before all, I would recommend the authors to clarify better what they mean by complexity (model complexity?), moreover that the word is a part of the title. For the readers will be easier to understand  and compare the models if input parameters and output ones are structured in a schematic drawing. This  will probably emphasize presence or lack of complexity. Once was stated that GrasProg1.0 input parameters are temperature, radiation, rainfall and PAW soil group (Line 357) and after several lines (Line 381) these parameters are expressed differently.

It is not clear whether the treatment protocols are common for Germany and Denmark to obtain comparable results.  At least unification of the soil parameters will be expected. Thee is lacking information about the sampling in Denmark. I am not convinced that the parameters introduced by Germany and Denmark are uniform.

I do not understand how GrasProg1.0 was improved by legacy effect. In the GrasProg1.0 model drought legacy factor was set at 7 days without explanation why. The soil water content below 30% could last different period. Please, explain more clearly.

In the discussion section authors claim that the GrasProg1.0 can be used to predict the best harvesting time. The paragraph between lines 360-368 shows too much convention and it is not clear how this convention affects the model.

The reference list must be  improved. It should reflect the Instructions for Authors which is not the case in the manuscript.

Some technical corrections

Line 21: …for separate cuts were…

Line 32: Please change Eurostat 2020 with [1] and renumber the references accordingly; The Latin names of the plants should be in italic.

Line 54, 162, 165, 166: change draught to drought

Line 85: what does 3999 mean?

Lines 100-102; 148-149: Please include date of accession of the mentioned sites.

line 117: abbreviations in the Table 1 title should coincide with the column captions

Line 124: unify expression N ha-1

Line 128: Please explain why sample area ranged of 10 to 12 m2; explain also how was obtained the dry mass (air dry or absolute dry)

Line 143; 314: please include properly the reference

Line 179: Include the reference Cullen et al.  2008 in the numbered list

Author Response

REVIEWER 2

The models in agriculture are useful tools in order to manage grassland ecosystems effectively. They are also important for predictions about the development of grass systems in time and space. This article discusses the ability of two modelling software to do so. Both are applied in the practice, but it seems that the authors need to prove that the one is better for certain applications. The study is of interest but by my opinion it needs some clarifications and improvements. Before all, I would recommend the authors to clarify better what they mean by complexity (model complexity?), moreover that the word is a part of the title. For the readers will be easier to understand  and compare the models if input parameters and output ones are structured in a schematic drawing. This  will probably emphasize presence or lack of complexity.

As a schematic representation of all the many processes and model input and output parameters is not feasible due to the enormous number, we have included a paragraph addressing the difference in complexity.

Once was stated that GrasProg1.0 input parameters are temperature, radiation, rainfall and PAW soil group (Line 357) and after several lines (Line 381) these parameters are expressed differently.

These two statements have now been unified.

It is not clear whether the treatment protocols are common for Germany and Denmark to obtain comparable results.  At least unification of the soil parameters will be expected. There is lacking information about the sampling in Denmark. I am not convinced that the parameters introduced by Germany and Denmark are uniform.

We have included information on the sampling in Denmark.  Regarding the parameters we believe that this is the grouping of soils regarding PAW, which was based on the classification systems.  As noted in the manuscript in several places these are not site-specific soil descriptions but general ones.

I do not understand how GrasProg1.0 was improved by legacy effect. In the GrasProg1.0 model drought legacy factor was set at 7 days without explanation why. The soil water content below 30% could last different period. Please, explain more clearly.

We have explained in more detail the physiological effect of a drought on subsequent grass growth.  We acknowledge that the drought period could vary, but with the limited data on such conditions we have simply set this effect according to the provided values.  We also state that further investigation and refinement of this is required.

In the discussion section authors claim that the GrasProg1.0 can be used to predict the best harvesting time. The paragraph between lines 360-368 shows too much convention and it is not clear how this convention affects the model.

The discussion part has been rewritten, and we hope that this is clearer now.

The reference list must be improved. It should reflect the Instructions for Authors which is not the case in the manuscript.

We have corrected the references which were not numbered.  However otherwise the instructions of the journal state that: Your references may be in any style, provided that you use the consistent formatting throughout.

Some technical corrections

These minor issues have been addressed

Line 21: …for separate cuts were…

Line 32: Please change Eurostat 2020 with [1] and renumber the references accordingly; The Latin names of the plants should be in italic.

Line 54, 162, 165, 166: change draught to drought

Line 85: what does 3999 mean?

Lines 100-102; 148-149: Please include date of accession of the mentioned sites.

line 117: abbreviations in the Table 1 title should coincide with the column captions

Line 124: unify expression N ha-1

Line 128: Please explain why sample area ranged of 10 to 12 m2; explain also how was obtained the dry mass (air dry or absolute dry)

Line 143; 314: please include properly the reference

Line 179: Include the reference Cullen et al.  2008 in the numbered list

Reviewer 3 Report

The MS entitled ‘How much complexity is required for modelling grassland production at regional scales?’ authored by Vogeler et al. evaluated growth models of GrasProg1.0 and APSIM using different complexity and input requirements from variety trials of perennial ryegrass. The MS falls well within the scope and aims of the journal and content of the MS may be of interest to wide readers of Land. However, there are few critical deficiencies which need clarification for reader’s convenience and imparting scientific soundness.

Title, it is perhaps better to replace with a conclusive statement for attracting reader’s attention.

Abstract, I shall strongly suggest adding a concrete problem statement as the very first phrase of abstract which has necessitated conducting this comparative study of growth models.

Meteorological features description is not making sense here and may be omitted.

Input requirements/variables on the basis of which models testing were performed need clarity.

‘The objective was to investigate if the simple model performs equally well’ needs clarity.

Response variables must be described in single phrase before stating results.

Results need to be described with respect to all response variables by objectively comparing working efficiency of both models.

‘These results show that without site-specific descriptions, the simple GrasProg1.0 model can be incorporated into a simple decision support tool for optimising pasture cutting regimes and management’ on which grounds authors have preferred GrasProg model over APSIM, it needs clarity.

Introduction, ‘Grasslands play a crucial role in livestock systems by providing forage for ruminant animals, with 80% of the global dairy milk, and 70% of the beef and veal produced from temperate grasslands’ must be rephrased.

In introduction, there are too many generalized statements which may be omitted such as However, temperature-limited herbage growth in spring and autumn, and moisture-limited growth in summer can result in feed deficits in intensively managed systems. This is commonly overcome by the use of supplemental feeding or forage crop blocks, which increases the costs of production’.

My prime concern is authors have not established study rationale by critically analysing peer-findings on comparative functioning of different models. Moreover, this section remained unacceptable without appropriately highlighting the research and knowledge gaps on the subject matter. Furthermore, technical information of  ryegrass (Lolium perenne) grasslands and their characteristics need to be briefly described.

Methodology, generalized statements like The ‘GrasProg model is a semi-mechanistic model for simulating grass growth for intensively managed ryegrass (Lolium perenne) dominated swards with typical non limiting N fertilization rates. The model only requires a few input parameters’ do not provide meaningful information here rather objective comparison of growth models based on peer-findings must be done in introduction sections for highlighting research gaps and study needs. Same is the case with APSIM model.

Authors have not provided any citations on which they designed statistical analysis.

Results have not been presented objectively and do not offer an explicit comparison of both models, so these must be re-written by describing per cent difference of both models for different response variables.

Perhaps manuscript has been compiled in rush as interpretation and discussion has also been added in result section such as ‘The defoliation frequency influences various plant traits, such as ‘tiller density, which greatly influence grass growth [50], and this might explain the disparities between measurements and simulations’ while a separate discussion section is also present.  

Discussion section must be enriched by adding interpretation of recorded findings for functioning of growth models and more recent works may be added to support or contradict the recorded findings.

Conclusion is highly generalized while authors need to amend it as per recorded findings, particularly describe research findings which prove superiority of GrasProg model over APSIM model. Moreover, authors must highlight perspectives of the research findings and future research needs.

Overall, proof-reading is recommended to rectify spelling and formatting issues throughout the text of ms.

Author Response

The MS entitled ‘How much complexity is required for modelling grassland production at regional scales?’ authored by Vogeler et al. evaluated growth models of GrasProg1.0 and APSIM using different complexity and input requirements from variety trials of perennial ryegrass. The MS falls well within the scope and aims of the journal and content of the MS may be of interest to wide readers of Land. However, there are few critical deficiencies which need clarification for reader’s convenience and imparting scientific soundness.

Title, it is perhaps better to replace with a conclusive statement for attracting reader’s attention.

We believe that the question is appropriate, as this finding might not be general to all environments, but would change it if the other reviewers or the editor are of the same opinion.

Abstract, I shall strongly suggest adding a concrete problem statement as the very first phrase of abstract which has necessitated conducting this comparative study of growth models.

A problem statement has been included

Meteorological features description is not making sense here and may be omitted.

Meteorological data are provided to emphasize the meteorological differences between sites in temperate region from Denmark to Southern Germany

Input requirements/variables on the basis of which models testing were performed need clarity.

Input variables for the models are now provided.

‘The objective was to investigate if the simple model performs equally well’ needs clarity.

Has been changed to 'The objective was to compare statistical model performance of the models under given low number of inputs, namely climate and PAW group. '

Response variables must be described in single phrase before stating results.

The response variables (biomass from individual cuts and annual total amounts) are now stated.

Results need to be described with respect to all response variables by objectively comparing working efficiency of both models.

The response variable have been evaluated for both models, using RMSE, NSE and R2 (as stated in the statistical analysis)

‘These results show that without site-specific descriptions, the simple GrasProg1.0 model can be incorporated into a simple decision support tool for optimising pasture cutting regimes and management’ on which grounds authors have preferred GrasProg model over APSIM, it needs clarity.

This sentence has been rephrased and should now be clearer in conjunction with the added first sentence of the abstract

Introduction, ‘Grasslands play a crucial role in livestock systems by providing forage for ruminant animals, with 80% of the global dairy milk, and 70% of the beef and veal produced from temperate grasslands’ must be rephrased.

The sentence has been deleted as it is not really relevant for the study

In introduction, there are too many generalized statements which may be omitted such as However, temperature-limited herbage growth in spring and autumn, and moisture-limited growth in summer can result in feed deficits in intensively managed systems. This is commonly overcome by the use of supplemental feeding or forage crop blocks, which increases the costs of production’.

The introduction has been rephrased and various statements have been deleted.

My prime concern is authors have not established study rationale by critically analysing peer-findings on comparative functioning of different models. Moreover, this section remained unacceptable without appropriately highlighting the research and knowledge gaps on the subject matter. Furthermore, technical information of  ryegrass (Lolium perenne) grasslands and their characteristics need to be briefly described.

The introduction has been re-phrased, further knowledge gaps have been included.  Information of ryegrass characteristics has been added.

Methodology, generalized statements like The ‘GrasProg model is a semi-mechanistic model for simulating grass growth for intensively managed ryegrass (Lolium perenne) dominated swards with typical non limiting N fertilization rates. The model only requires a few input parameters’ do not provide meaningful information here rather objective comparison of growth models based on peer-findings must be done in introduction sections for highlighting research gaps and study needs. Same is the case with APSIM model.

We have included more information about the input parameters and processes considered in the tow different models, and hope that the difference in complexity is clearer now.

Authors have not provided any citations on which they designed statistical analysis.

References to R statistics packages are provided. 

Results have not been presented objectively and do not offer an explicit comparison of both models, so these must be re-written by describing per cent difference of both models for different response variables.

Perhaps manuscript has been compiled in rush as interpretation and discussion has also been added in result section such as ‘The defoliation frequency influences various plant traits, such as ‘tiller density, which greatly influence grass growth [50], and this might explain the disparities between measurements and simulations’ while a separate discussion section is also present. 

This paragraph has been shifted to the discussion

Discussion section must be enriched by adding interpretation of recorded findings for functioning of growth models and more recent works may be added to support or contradict the recorded findings.

We have included the main finding, that GrasProg shows a similar fit between measurements and predictions as APSIM, and is thus a suitable grass growth model for North-west European conditions, especially regarding the much lower input requirement for the model. We also included similar findings from the few other studies that we are aware of.

Conclusion is highly generalized while authors need to amend it as per recorded findings, particularly describe research findings which prove superiority of GrasProg model over APSIM model. Moreover, authors must highlight perspectives of the research findings and future research needs.

We have limited our Conclusions to the main finding of our paper.  As this section is not mandatory for the journal we believe that this should only show these.

Round 2

Reviewer 2 Report

The authors have addressed all my suggestions and have answered to the comments. The text has been improved accordingly. I have no more comments to this article.

Reviewer 3 Report

I am pleased to confirm authors have significantly improved the manuscript except not agreeing to amending the title (that's still okay for me). The Ms might be accepted in current form.